# Artificial Intelligence of Things Applied to Assistive Technology: A Systematic Literature Review

**DOI:** 10.3390/s22218531

**Published:** 2022-11-05

**Authors:** Maurício Pasetto de Freitas, Vinícius Aquino Piai, Ricardo Heffel Farias, Anita M. R. Fernandes, Anubis Graciela de Moraes Rossetto, Valderi Reis Quietinho Leithardt

**Affiliations:** 1School of Sea, Science and Technology, University of the Itajaí Valley, Itajaí 88302-901, Brazil; 2Federal Institute of Education, Science and Technology Sul-Rio-Grandense, Passo Fundo 99064-440, Brazil; 3COPELABS, Lusófona University of Humanities and Technologies, Campo Grande 376, 1749-024 Lisboa, Portugal; 4VALORIZA, Research Center for Endogenous Resources Valorization, Instituto Politécnico de Portalegre, 7300-555 Portalegre, Portugal

**Keywords:** AIoT, artificial intelligence, assistive technology, deep learning, machine learning

## Abstract

According to the World Health Organization, about 15% of the world’s population has some form of disability. Assistive Technology, in this context, contributes directly to the overcoming of difficulties encountered by people with disabilities in their daily lives, allowing them to receive education and become part of the labor market and society in a worthy manner. Assistive Technology has made great advances in its integration with Artificial Intelligence of Things (AIoT) devices. AIoT processes and analyzes the large amount of data generated by Internet of Things (IoT) devices and applies Artificial Intelligence models, specifically, machine learning, to discover patterns for generating insights and assisting in decision making. Based on a systematic literature review, this article aims to identify the machine-learning models used across different research on Artificial Intelligence of Things applied to Assistive Technology. The survey of the topics approached in this article also highlights the context of such research, their application, the IoT devices used, and gaps and opportunities for further development. The survey results show that 50% of the analyzed research address visual impairment, and, for this reason, most of the topics cover issues related to computational vision. Portable devices, wearables, and smartphones constitute the majority of IoT devices. Deep neural networks represent 81% of the machine-learning models applied in the reviewed research.

## 1. Introduction

According to the World Health Organization, about 15% of the world population has some type of disability, totaling approximately 190 million people [1]. This same source also indicates this number continues to grow due to the increase in chronic health conditions and the aging of the world population. Taking into consideration that disabilities may be of a temporary or permanent nature, they may encompass a wide range of special needs, restrictions, and health conditions. These disabilities can be represented by degenerative diseases such as Parkinson’s, amyotrophic lateral sclerosis (ALS), and Alzheimer’s, physical, mental, visual, and hearing disabilities, and chronic non-communicable diseases. Additionally, conditions can result from aging, a period marked by the highest rates of the development of some disabilities [1].

According to the United Nations Convention on the Rights of Persons with Disabilities (CRPD), disability is not an attribute of the person but is the result of environmental and behavioral barriers that arise from the interaction between people with disabilities and society, whereby preventing them from participating equally, fully, and effectively as citizens in society. Therefore, dealing with the obstacles that affect people with disabilities contributes to the improvement of their social participation in general. Within this context, Assistive Technology (AT) contributes directly to the reduction of difficulties encountered by people with disabilities in their daily lives. Assistive Technology allows them to live independently, healthy, and productively, to receive education and be participants in the labor market and society in a worthy manner [2].

Assistive Technology encompasses services, products, methodologies, strategies, and practices that aim to minimize and/or eliminate restrictions and limitations imposed on a person due to a disability or incapacity [3]. It focuses on providing independence, quality of life, and social inclusion to people with disabilities. Some examples of Assistive Technology are hearing aids, memory aids, eyeglasses, wheelchairs, pill organizers, and communication aids. Assistive Technology has made great advances in its integration with Artificial Intelligence of Things—from now on referred to as AIoT—devices and machine learning [4,5,6]. AIoT processes and analyzes the large amount of data generated by Internet of Things—from now on referred to as IoT—devices and applies Artificial Intelligence techniques, specifically, machine learning, to discover patterns for generating insights and assisting in decision making [7].

When applied to AT, AIoT allows the conception of an array of disruptive solutions to address the disability issue. Some examples of such solutions are navigation systems for blind people, voice assistants for people with disabilities [8], the remote monitoring of health conditions [9], telemedicine and telehealth [10], communication systems based on sign language [11], auxiliary memory for people with cognitive disabilities, and a series of smart objects such as medicine dispensers, wheelchairs [12], exoskeletons [13], etc. These are some of the numerous applications of great value for those in need, quoting Mary Pat Radabaugh (the former director of IBM’s National Support Center for People with Disabilities in 1988) “For people without disabilities, technology makes life easier. For people with disabilities, technology makes life possible” [14].

Given the importance of the development of AIoT applied to Assistive Technology, this systematic literature review aims to identify the machine-learning models used in the different research on this topic. Since the reviews of the literature on the proposed subject are still small, when compared to the amount of relevant research conducted in the area, it becomes important that the data and evidence collected through recent studies are presented in a coherent and cohesive fashion. The present article collects previous research to present such a review.

For this objective the present Systematic Literature Review (SLR) is based on guidelines defined by Kitchenham and Charters in 2007 [15] and by Petersen et al. [16]. In this scope, an SLR carries out a survey of the relevant previous research conducted on a particular theme or research question to find evidence that it answered its proposed objectives. An SLR is then considered evidence-based research. This process takes place through a rigorous, reliable, and replicable methodology [15].

The idea of using evidence-based research in the field of computer science was originally proposed by Kitchenham, Dybå, and Magne Jørgensen in 2004 and 2005 [17,18], where researchers used an “Evidence-based Software Engineering” (EBSE) methodology in the research and practice of software engineering. The stages of a systematic literature review that were addressed in this work were: the planning, consisting of the creation of the SLR protocol; the conduction, consisting of the application of the SRL protocol; and the SLR documentation and report [19]. Each of these steps is composed of a series of steps, as defined, and presented through this SLR.

The survey of the topics approached in this article also highlights the context of such research, their application, the IoT devices used, and gaps and opportunities for further development. This article is organized as follows: Section 2 presents the concepts related to the research. Section 3 presents the research methodology used, which includes the research’s objective and questions, the steps of the systematic literature review, and the threats to the validity of the results. Section 4 presents the research results of this review. Section 5 presents the conclusion based on the study results and recommendations for future studies.

## 2. Assisted Technology, AIoT, and Machine Learning

Aiming at minimizing and/or eliminating restrictions and limitations imposed on a person due to a disability or incapacity, the World Health Organization defines Assistive Technology as a wider term encompassing any system or service related to assistive products and services. The Assistive Technology Industry Association defines Assistive Technology’s products and services as any item, piece of equipment, hardware, or software intended to assist people with some type of disability. Such products and services are a result of the combination of AT and AIoT [11]. Industry 4.0 is an ally for the improvement of AT. The development of devices based on Artificial Intelligence and Internet of Things, at a low cost, will benefit a large part of society that depends on ATs for a better living condition. On the other hand, persons with disabilities can qualify for the labor market in the Industry 4.0 environment by using ATs [20]. The design and forms of Assistive Technologies, whose task is to enable a greater involvement of people with disabilities in the field of employment, are extremely important.

Artificial Intelligence of Things is obtained by the combination of IoT [20] and Artificial Intelligence (AI) techniques [21]. IoT is defined as any device able to interconnect—such as sensors—and can collect relevant data in real time [22]. This relevance is revealed by processing the obtained data through Artificial Intelligence models, especially making use of machine learning (ML). Some cases also require the usage of deep learning (DL) to analyze the collected data to extract useful information for decision making [23,24,25]. The application of ML techniques shows promise for the healthcare sector [26] by improving efficiency in this sector [27].

The term AI was coined by John McCarthy [28], who is considered the father of AI, in 1956, during the first AI conference at Dartmouth College. McCarthy defines AI as: It is the science and engineering of making intelligent machines, especially intelligent computer programs. It is related to the similar task of using computers to understand human intelligence, but AI does not have to confine itself to methods that are biologically observable [29]. Several areas have been expanding faster, recently, and need dynamic solutions that can be solved with AI [30], such as sustainability [31], health [32], telecommunication systems [33], data privacy [34,35], electric vehicles [36], and electrical power systems [37,38,39].

Alan Turing, however, proposed, in 1950, the question: “Can machines think?”. The Turing test was then launched with the aim of determining whether a computer can demonstrate the same intelligence as a human being [40]. To pass this test, the system would need to possess capabilities that are currently the subject of study in machine learning, such as natural language processing [41], knowledge representation [42], and automated reasoning [43]. Given the advances in AI models, several applications are being used to improve the quality of life of people with physical disabilities and improve applications for smart healthcare [44], such as using smart robots [45,46,47], or more specific applications, such as in sign language [48,49,50,51,52,53].

Machine learning is a subfield of AI that aims at developing models and computer programs that can learn automatically by extracting knowledge from data [54]. These programs must be able to improve and extend themselves from experience, without being explicitly programmed. In the IoT context, these models are used to process and analyze a large amount of data collected by devices, automatically discover patterns, and generate meaningful insights from this data. Such a task would be impossible for humans to perform manually. Some of these ML models are echo state networks [55], ensemble learning methods [56,57,58], k-nearest neighbors (K-NN) [59], group method of data handling (GMDH) [60], long short-term memory (LSTM) [61], convolutional neural networks (CNNs) [62,63,64,65], and adaptive neuro-fuzzy inference system (ANFIS) [66].

Deep learning is a subfield of machine learning. Deep learning specifically studies deep neural networks. Like ML, deep learning also uses data-based learning methods; however, computation and processing are completed using multilayer neural networks [67]. The experimental results and heuristic considerations suggest that deep architectures are more suitable than shallow ones for modern applications facing very complex problems, e.g., vision and human language understanding and the processing of big data. [68]. These models can be applied in prediction [69,70,71], classification [72,73,74], and optimization [75,76,77] problems.

## 3. Research Methodology

A systematic literature review (SLR) was performed to achieve the objectives of the current study. SLR is a methodological review of research results that aims to aggregate existing evidence on a research problem, as well as identify, select, evaluate, and summarize primary articles considered relevant on the research topic in an unbiased and repeatable way. SLR is considered a secondary study for aggregating previous studies [15].

The stages and sub-stages of a systematic review addressed in this work were: planning, conducting, and documenting the review. After presenting the research objectives and questions, the topics related to planning were displayed according to the progress of the SLR. This approach was based on the work presented by Kitchenham et al. [78], Banijamali and Ahmad et. al. [79]. Banijamali and Ahmad et. al. [79] define these research stages as: the planning, where this step includes the activities of identifying the research objective, defining the research questions, developing a review protocol, and evaluating the review protocol, during which these activities can be done interactively; the conducting, where this step includes the activities of identifying primary articles using search strategies, selecting studies using inclusion and exclusion selection criteria for studies, extracting the data, and synthesizing the data; the publication, where this step includes the activities of specifying the report, formatting the report, and evaluating the report.

Figure 1 presents the systematic review steps (guidelines) based on the proposed by Kitchenham [80].

Figure 2 presents the systematic review process model on which this work based. The process was based on the proposed by Kitchenham [80].

### 3.1. Research Purpose

The objective of this literature review is to identify the models of ML used in the research of Artificial Intelligence of Things-based Assistive Technology. Additionally, it aims to identify the context of these models’ applications through the survey of the topics of study, the IoT devices used, and to identify development gaps and opportunities. Due to the great magnitude of conditions characterized as disabilities, the scope of this study was defined for research that addresses the following disabilities: visual impairment, hearing impairment, cognitive impairment, and degenerative diseases such as ALS, Alzheimer’s, and Parkinson’s. Table 1 presents the research questions of this study.

QP1 intends to identify the machine-learning models applied in the development of AIoT applied to Assistive Technology. All models mentioned in the selected primary articles were identified for this purpose.

QP2 intends to provide a context related to the topics addressed in the development of AIoT applied to Assistance Technology. The study topics addressed by the selected primary articles were identified for this purpose.

QP3 intends to provide context related to the types of IoT devices that have been used for the development of AIoT applied to Assistance Technology. All IoT devices listed in the proposed solutions were identified for this analysis. The devices were also classified as Arduino, RaspberryPY, or Nvidia Jetson-based.

QP4 identifies the gaps for AIoT applied to Assistive Technology research and development, indicating for which disability or incapacity the study intends to develop a solution.

### 3.2. Research Process

This section aims to detail the research process used in this SLR. A database search or automatic search strategy was used for this study. It consists of searching digital libraries using a search string [80]. A search string consists of combining keywords using logical operators such as OR and AND. Usually, synonyms of each keyword are grouped by the OR operator, and an AND operation is used to join these groups.

The keywords defined for this study were: Assistive Technology, AIoT, and Machine Learning. Interactive tests were carried out on each of the databases to identify the search strings whose returns were the most significant in terms of scope and relevance of the studies. The selected libraries are justified for being among the most used in research in computer science [80]. Table 2 presents the databases used and the respective search strings defined after the iterative validation process.

The automatic database search was performed on 14 September 2021. Each database returned a set of articles, as shown in Table 3. These articles were processed to remove duplicated entries; this task was performed automatically by the tool Parsifal. Initially, a total of two hundred and sixty-seven articles were selected, of which seventy-nine were duplicates, leaving one hundred and eighty-eight articles. The set of articles resulting from the research stage passed to the next stage of the SLR, the Study Selection.

### 3.3. Study Selection Criteria

This section aims at detailing the research process used in this SLR. A database search or automatic search strategy was used for this study. It consists of searching digital libraries using a search string [80]. A search string consists of combining keywords using logical operators such as OR and AND. Usually, synonyms of each keyword are grouped by the OR operator, and an AND operation is used to join these groups.

This section aims to present the selection and exclusion criteria used in this SLR. The presented criteria were preconditions for the acceptance or exclusion of an article in the SLR. This selection seeks to identify relevant studies that can answer the proposed research questions [78,79,80]. The criteria were applied to the set of articles resulting from the search process. Some of the criteria were applied directly to the databases, according to the available filters, with variations for each of the databases. Table 4 presents the inclusion criteria, and the indication of those applied directly to the bases. Table 5 lists the exclusion criteria.

It is worth mentioning that it was enough that if one of the exclusion criteria was met, the article was excluded. On the other hand, for an article to be selected, it was necessary that all the inclusion criteria were satisfied. Initially, the title and abstract of each article were read and taken into consideration when applying the selection criteria and, in cases where the reading was insufficient for this, the article was read until the selection or exclusion could be confirmed. At the end of this stage, a total of 30 articles remained; this set passed to the next stage of the process, the Quality Assessment.

### 3.4. Quality Assessment

This section presents the quality assessment process used in this SLR. The previous step resulted in a set of pre-selected articles. However, for these articles to be considered accepted by this SLR, it was necessary to go through the quality assessment process. At this stage, each study was evaluated to ensure the quality of the data that was extracted in the subsequent data extraction stage [15,80], meaning that it intended to identify studies that were relevant to answer the research questions.

Table 6 shows a questionnaire containing five questions prepared for this SLR to assess the quality of the articles. Each question has three options as answers and each option has its respective score. Options are “yes”, “partially”, and “no”, scoring 1.0, 0.5, and 0, respectively. The total score for each article was defined by the sum of the values obtained from five answers. A maximum value of 5.0 indicated a well-matched article for this SLR and a minimum of 0.0 indicates that the article was not suitable for this SLR. A cutoff score of 2.0 was defined, so only articles with a score greater than 2.0 were effectively considered as accepted for this SLR. This quality assessment method was implemented and applied using Parsifal.

The primary articles were read in their entirety for the attribution of grades. Table 7 shows the result of the quality assessment. The input set of this stage contained 30 articles, of which 3 articles received a score less than or equal to 2.0. Thus, 27 articles passed to the next stage, the Data Extraction. The selected studies were submitted to the next stage of the SLR, the Data Extraction.

### 3.5. Data Extraction

The data extraction process makes use of a data extraction form generated, specifically for this SLR, using the Parsifal tool. Filling in the fields of this form after the reading of each selected article allows the recovery of data to answer the research questions raised by this SLR, as can be seen in Table 1. This form also collects metadata used to identify the studies individually, thereby assisting the extraction process [78]. Table 8 presents the fields and the purpose of each field within the extraction form.

Table 8 brings ten data properties defined for this study where PD1 to PD3 were used to identify and locate articles. The other properties were defined to answer the research questions where PD4 answers QP1; PD5 and PD6 answer QP2; PD7 answers QP3; and PD8 answers QP4. Data was extracted and organized by the Parsifal tool after the complete reading of each of the selected articles (see Table 7), thereby facilitating the extraction process.

### 3.6. Threats to the Validity of the Study

Biases in the identification of primary articles and in the extraction of data from the articles, were corroborated by the fact that each researcher was responsible for evaluating a set of disjointed articles, in the selection, as well as in the quality assessment, with no peer validation, is a threat to the validity of the study. Another threat is the small number of articles selected in this SLR, which implies the possibility that the sample is not representative to extract evidence that can effectively answer the research questions.

The selection of databases, or digital libraries, can also be considered a threat, since these may not cover the completeness of the studies carried out in the context of the problem, which includes questions from different areas of sociology and medicine.

## 4. Results

This section summarizes the findings and results of the analyses of the selected primary articles. The selection process conducted in the selected databases collected an initial amount of two hundred and sixty-seven articles. Of these articles, twenty-six were considered for this SLR (see Table 7) for their contribution to the topic. Section 4.1 presents the contribution of the articles with a score greater than 3 (see Table 7). Section 4.2 proposes answers to the research questions presented in Table 1, which were based on the data extracted by the form presented in Table 8. Section 4.3 aims at summarizing the research findings.

### 4.1. Contributions of Selected Articles

The articles presented below have relevant contributions to the research in AIoT applied to Assistive Technology. It can be noticed that visual impairment is the area of research with the highest number of presented works.

The work of Junior et al. [81] present a framework that applies computer vision and machine-learning techniques, through the IoT network with the use of cloud computing, for a capacity increase. The images are captured by an IoT device and sent to an edge element (IoT node), which processes them, identifies objects, computes distances, and, ultimately, converts that information into audible commands to provide guidance for visually impaired people.

Chang et al. [82] present a deep-learning-based wearable medicines recognition system for visually impaired people. The proposed system is composed of a pair of wearable smart glasses, a wearable waist-mounted drug pills recognition device, a mobile device application, and a cloud-based management platform. This system uses deep-learning technology to assist visually impaired people in identifying drug pills and help them avoid taking the wrong medicines. The experimental results show that the accuracy of the proposed system reached up to 90%.

A system for localized scene understanding to assist sufferers of visual disabilities is proposed by Ghazal et al. [83]. The system determines the user’s indoor location using Wi-Fi fingerprinting and synthesizes a real-time description of the surrounding environment, using deep learning, and sensory information collected from IoT sensors. The results show that the system can be an effective tool in helping the visually challenged navigate unknown environments by using increasingly available smart home technologies.

Chang et al. [84] present a wearable smart glasses-based drug pill recognition system, using deep learning, for visually impaired people to improve their medication-use safety. The system consists of a pair of wearable smart glasses, an artificial intelligence-based intelligent drug pill recognition box, a mobile device app, and a cloud-based information management platform. The experimental results show that a recognition accuracy of up to 95.1% can be achieved.

A wearable assistive system based on artificial intelligence edge computing techniques to help visually impaired consumers safely use marked crosswalks, or zebra crossings, is presented by Chang et al. [86]. The system consists of a pair of smart sunglasses, a waist-mounted intelligent device, and an intelligent walking cane (stick). A deep-learning technique is adopted for zebra crossing image recognition in real time. The experimental results show that the accuracy of the real-time zebra crossing recognition of the proposed system can reach up to 90%.

In the study of Su et al. [88], we find the design of a finger-worn device that can be applied by visually impaired users for recognizing traditional Chinese characters on a micro IoT processor. The system on the index finger contains a small camera and buttons, which capture images by identifying the relative position of the index finger to the printed text, and the buttons are applied to capture an image by visually impaired users and provide the audio output of the corresponding Chinese character by a voice prompt. To recognize Chinese characters, English letters, and numbers, a robust Chinese optical character recognition (OCR) system was developed according to the training strategy of an augmented convolution neural network algorithm. The Chinese OCR system can segment a single character from the captured image, and the system can accurately recognize rotated Chinese characters. The experimental results revealed that compared with the OCR application programming interfaces of Google and Microsoft, the proposed OCR system obtained a 95% accuracy rate in dealing with rotated character images, whereas the Google and Microsoft OCR APIs only obtained 65% and 34% accuracy rates.

Yadav et al. [89] propose a smart navigation system that relentlessly scans the environment, and then detects and classifies neighboring objects using a 4-layered convolutional neural network (CNN), which has been trained on a data set containing 2513 permutations of the various images of household objects that an individual may encounter in daily life. The device has achieved a success rate of serving within a response time of less than 50 ms. The accuracy of the CNN algorithm at 94.6% also sets a distinguished benchmark as an object detection algorithm, thereby contributing to the success of the simulations of the proposed device in a constrained environment.

Kandoth et al. [96] present an application developed based on a smart portable cane equipped with sensors and a camera to help the visually impaired to understand their surroundings, detect obstacles and avoid them, using computer vision, neural networks, and IoT in outdoor spaces, thereby helping them to navigate safely. The idea is to implement the YOLO (you-only-look-once) algorithm for obstacle detection using the DarkNet 1.0 framework, which is then used for object avoidance using an ultrasonic sensor. The results are considerably better than those of the SegNet networks when trained only with the original images and other state-of-the-art results using the Synthia database.

Karkar et al. [105] present the concept of scene to speech (STS). STS recognizes the elements in a captured image or a video clip and speaks, loudly, informative textual content that describes the scene. The contemporary progression of convolution neural networks allows attaining object recognition procedures, in real time, on mobile handheld devices.

Boppana et al. [93] present the design of a prototype of an assistive device for deaf–mute people to reduce the communication gap with normal people. This device is portable and can hang over the neck. This device allows the person to communicate with sign hand postures to recognize different gesture-based signs. The controller of this assistive device was developed—for processing the images of gestures—by employing various image-processing techniques and deep-learning models to recognize the sign. This sign is converted into speech in real time using a text-to-speech module. This sign language converter was found to be 99% accurate in recognizing the signs and generating the correct words.

The study of Lee et al. [101] present the design and implementation of a smart wearable American Sign Language interpretation (ASL) system, using deep learning, which applies sensor fusion that “fuses” six inertial measurement units (IMUs). The IMUs are attached to all fingertips and the back of the hand to recognize sign language gestures; thus, the proposed method is not restricted by the field of view. The model achieves an average recognition rate of 99.81% for dynamic ASL gestures. The ASL recognition system can be integrated with ICT and IoT technology to provide a feasible solution to assist hearing-impaired people in communicating with others.

The paper of Javed and Sarwar [95] proposes a smartphone-based end-to-end novel framework, named PP-SPA, for privacy-preserved human activity recognition (HAR) and real-time activity functioning support using a smartphone-based virtual personal assistant. PP-SPA helps to improve the routine life functioning of cognitive-impaired individuals. It uses a highly accurate machine-learning model that takes input from smartphone sensors (i.e., accelerometer, gyroscope, magnetometer, and GPS) for accurate HAR and uses a digital diary to recommend real-time support for the improvement of an individual’s health. PP-SPA yields a proficient accuracy of 90% with the Hoeffding tree and logistic regression algorithm, which endeavors reasonable models in terms of uncertainty.

The paper of Jacob et al. [13] presents an artificial intelligence-powered smart and light weight exoskeleton system (AI-IoT-SES), which receives data from various sensors, classifies them intelligently, and generates the desired commands via IoT for rendering rehabilitation and support, with the help of caretakers, for paralyzed patients in smart and connected communities. The navigation module uses AI-and IoT-enabled simultaneous localization and mapping. The casualties of a paralyzed person are reduced by commissioning the IoT platform to exchange data from the intelligent sensors with the remote location of the caretaker monitoring the real-time movement and navigation of the exoskeleton. The experimental results simulated show that the proposed system is an ideal method for rendering rehabilitation and support for paralyzed patients in smart communities.

Al Shabibi and Kesavan [94] present a cost-effective smart wheelchair-based Arduino Nano microcontroller and IoT technology that have several features to benefit disabled people, especially poor people, who cannot afford expensive smart wheelchairs or the required help to finish daily life tasks without external help. The smart wheelchair, which is affordable to a wide range of disabled people and is based on the Arduino Nano, is equipped with a module to give Wi-Fi access, and another to detect a fall with voice message notification using an IFTTT platform, obstacle detection with a buzzer, LEDs to work as hazard lights, a voice recognition system, and joysticks to control the wheelchair.

Wang et al. [99] present the development of a compact, non-obtrusive and ergonomic wearable device to measure signals associated with human physiological gestures, and, thereafter, generate useful commands to interact with the environment. It uses machine learning and non-invasive biosensors on the top of the ears to identify eye movements and facial expressions with over 95% accuracy. Users can control different applications, such as robots, powered wheelchairs, cell phones, smart home, or other IoT, devices. The experimental results show satisfactory performance in different applications.

Sharma et al. [97] present DeTrAs: a deep-learning-based Internet of Health framework for the assistance of Alzheimer’s patients. It works with three components: a recurrent neural network-based Alzheimer’s prediction scheme, which uses sensory movement data; an ensemble approach for the abnormality tracking of Alzheimer’s patients designed to comprise two parts: (a) a convolutional neural network-based emotion detection scheme and (b) a timestamp window-based natural language processing scheme; and finally, an IoT-based assistance mechanism for Alzheimer’s patients. The evaluation of DeTrAs depicts, almost, a 10–20% improvement in terms of accuracy in contrast to the different existing machine-learning algorithms.

### 4.2. Research Questions Answers

This section presents the answers to the four established research questions.

QP1. What are the Machine Learning models used in AIoT applied to Assistive Technology?

Out of the twenty-seven primary articles studied, 81% presented solutions based on ANN, 15% of them applied other ML techniques, and 7% did not present the techniques used. As shown in Table 9, within the context of neural networks, the following ML techniques were addressed: ANN, CNN, the use of multiple-CNN, clever CNN, R-CNN (region-based CNN), faster R-CNN, PNN (probabilistic neural network), RNN (recurrent neural network), and multi-trained DL models. The models not based on neural networks were: Hoeffding tree, logistic regression, naïve Bayes, random forest, k-means, linear regression, independent component analysis, support vector machines (SVM), and HOG (histogram of oriented gradients).

QP2. What are the topics of the study that have been researched in the context of AIoT applied to Assistive Technology?

Figure 3 presents a word cloud, created using the keywords of each of these articles, to provide a general idea about the topics being studied on the selected primary articles.

The research topics discussed in the primary articles were assisted locomotion, assisted navigation, facial recognition, human activity recognition, image captioning, object detection, object recognition, OCR (optical character recognition), scene to speech, self-balancing object, smart assistant, speech recognition, rehabilitation, text to speech, and text detection. Table 10 shows the articles wherein these topics occurred.

QP3. What are the IoT devices used in the context of AIoT applied to Assistive Technology?

The IoT devices used in the primary articles’ proposed solutions: portable devices, wearables, various sensors, smartphones, cane, finger worn wireless, exoskeletons, wheelchairs, and others. Table 11 presents the articles wherein these devices were used.

Out of the primary articles, 60% of them showed the use of RaspberryPY, Arduino, and Nvidia Jetson-based devices, at 41%, 15%, and 7%, respectively, of the device total. Table 12 presents the articles that used RaspberryPY, Arduino, and Nvidia Jetson-based devices in their research.

QP4. Is there a disparity in the number of studies found according to the problems selected in the research?

The selected articles point to a large disparity in the development of AIoT applied to Assistive Technology in relation to the impairments chosen for this study. A total of 52% of the total number of articles addressed issues related to visual impairment, 19% of them addressed issues related to hearing impairment, and the rest were distributed between 11% motor coordination impairments, 7% degenerative diseases, and 4% cognitive impairment. This shows that most AIoT applied to Assistive Technology developments, within the selected articles, address visual impairment.

### 4.3. Threats to the Validity of the Study

Biases in the identification of primary articles and in the extraction of data from the articles, were corroborated by the fact that each researcher was responsible for evaluating a set of disjointed articles, in the selection, as well as in the quality assessment, with no peer validation, is a threat to the validity of the study. Another threat is the small number of articles selected in this SLR, which implies the possibility that the sample is not representative to extract evidence that can effectively answer the research questions. The selection of databases, or digital libraries, can also be considered a threat, since these may not cover the completeness of the studies carried out in the context of the problem, which includes questions from different areas of sociology and medicine.

## 5. Conclusions

This article presented a systematic literature review, which aimed to identify machine-learning algorithms and techniques used in AIoT applied to Assistive Technology solutions, as well as the context of its applications. Two hundred and sixty-seven articles were pre-selected using automatic search mechanisms on previously selected databases or digital libraries. After applying the selection criteria and quality assessment process, twenty-seven articles were considered from this initial set.

The final set of articles was submitted to the data extraction process, whereafter extraction, the data was organized, summarized, and analyzed to answer the research questions raised in this SLR. After surveying the findings, it was possible to conclude that 50% of the analyzed articles addressed visual impairment, thereby identifying a gap and an opportunity to develop Assistive Technology for all other disabilities. It was also possible to observe that most topics were influenced by many studies focused on visual impairment, resulting in a majority of the topics related to, or based on, computer vision.

It was possible to identify that 81% of the studies used machine-learning algorithms and techniques based on neural networks and only 15% of the studies used different techniques. This shows not only the interest of researchers in neural networks but also the great applicability of these learning techniques in the solution of Assistive Technology problems. Conversely, it also shows that there is a gap waiting to be filled in relation to the other algorithms and techniques.

Some threats to the validity of the results were also raised, such as biases in the identification of primary articles and extraction of results, the selection of digital libraries, and the number of studies selected in the SLR. Future work, in relation to the first threat, should intend to adopt peer review in the SLR stages, such as the application of selection criteria, quality assessment, and data extraction. To handle the second threat, the inclusion of other digital libraries and databases not covered by this SLR can also significantly contribute to the scope of primary articles. Finally, for the last threat, the adoption of a hybrid search strategy, possibly using snowballing and a manual search, in addition to an automated search, should be considered in future work.

## Figures and Tables

**Figure 1 sensors-22-08531-f001:**
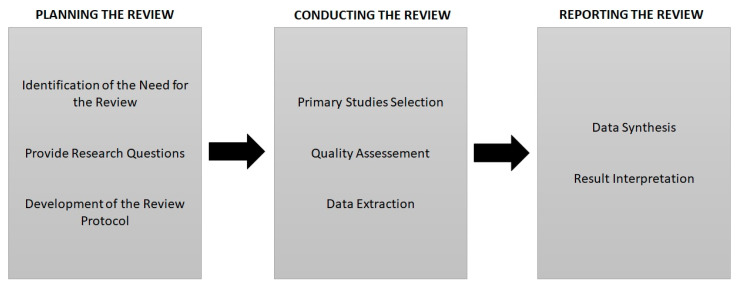
The systematic review steps (guidelines) [80].

**Figure 2 sensors-22-08531-f002:**
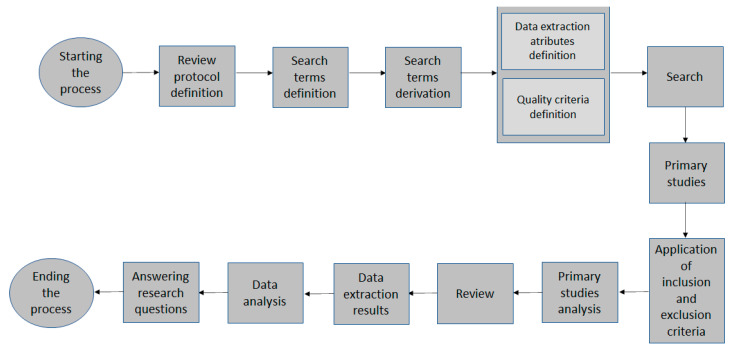
The systematic review model [80].

**Figure 3 sensors-22-08531-f003:**
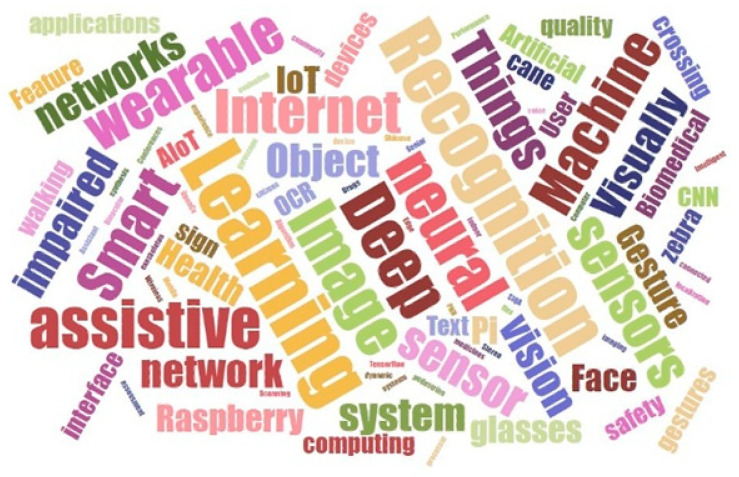
Word cloud.

**Table 1 sensors-22-08531-t001:** Research questions.

ID	Question	Justification
QP1	Which are the machine-learning models used in AIoT applied to Assistive Technology?	Identification of ML models used in AIoT applied to AT.
QP2	What are the topics of study that have been researched in the context of AIoT applied to Assistive Technology?	Providing context related to research topics.
QP3	What are the IoT devices used in the context of AIoT applied to Assistive Technology?	Providing context related to the types of IoT devices used.
QP4	Is there a disparity in the number of studies found according to the problems selected in the research?	Identify gaps for research and the development of solutions.

By the authors.

**Table 2 sensors-22-08531-t002:** Databases and search strings.

Data Base	ID	Search String	URL
El Compendex	EIC	(“assistive technology” OR “impaired people”) AND (“AIoT” OR “IoT” OR “internet of things”) AND (“machine learning” OR “deep learning” OR “neural networks”)	http://www.engineeringvillage.com (accessed on 4 August 2022)
IEEE Digital Library	IEEE	(“assistive technology" OR “impaired" OR “parkinson" OR “alzheimer") AND (“IoT" OR “AIoT" OR “Internet of Things" OR “artificial intelligence or things”) AND (“machine learning” OR “deep learning” OR “neural network”)	http://ieeexplore.ieee.org (accessed on 12 August 2022)
ISI Web of Science	WOS	(“assistive technology” OR “impaired” OR “parkinson” OR “alzheimer”) AND (“iot” OR “aiot” OR “Internet of Things” OR “artificial intelligence or things”) AND (“machine learning” OR “deep learning” OR “neural network”)	http://www.isiknowledge.com (accessed on 6 August 2022)
ScienceDirect	SCD	(“assistive technology”) AND (“IoT” OR “internet of things”) AND (“machine learning” OR “deep learning” OR “neural networks”)	http://www.sciencedirect.com (accessed on 10 August 2022)
Scopus	SCPS	(“assistive technology” OR “impaired” OR “parkinson” OR “alzheimer”) AND (“iot” OR “aiot” OR “Internet of Things” OR “artificial intelligence or things”) AND (“machine learning” OR “deep learning” OR “neural network”)	http://www.scopus.com (accessed on 15 August 2022)

By the authors.

**Table 3 sensors-22-08531-t003:** Number of selected articles from each database.

Data Base	Number of Selected Articles
El Compendex	37
IEEE Digital Library	63
ISI Web of Science	32
ScienceDirect	67
Scopus	68
Number of articles	267
Number of duplicated articles	79
Number of selected articles	188

By the authors.

**Table 4 sensors-22-08531-t004:** Inclusion criteria.

ID	Criteria	Applied Directly to the Databases
IC 1	Studies published between 2017 and 2021	EIC, IEEE, WOS, SCD, SCPS
IC 2	Peer-reviewed primary articles	EIC, WOS, SCD, SCPS
IC 3	Studies within the context of AIoT applied to AT, within the scope of deficiencies established	-
IC 4	Articles published in English	EIC, IEEE, WOS, SCPS

By the authors.

**Table 5 sensors-22-08531-t005:** Exclusion Criteria.

ID	Criteria	Applied Directly to the Databases
EC 1	Secondary or tertiary studies, studies within the context of AIoT applied to AT	-
EC 2	Studies within the context of AIoT applied to AT, within the scope of deficiencies established	-
EC3	Short articles, books, and gray literature (manuals, reports, theses, and dissertations)	EIC, WOS, SCD, SCPS
EC 4	Not having access to the study	-
EC 5	Duplicated study	-
EC 6	Redundant studies by the same author	-
EC 7	Studies prior to 2017	EIC, IEEE, WOS, SCD, SCPS

By the authors.

**Table 6 sensors-22-08531-t006:** Questionnaire for quality evaluation.

ID	Question
AQ 1	Are the study objectives clearly defined?
AQ 2	Is the problem to be solved clearly described?
AQ 3	Do the authors describe in detail the use of the ML models used in the solution?
AQ 4	Did the study perform a well-described experiment to evaluate the proposal?
AQ 5	Do the findings of the study indicate a validity relevant to it?

By the authors.

**Table 7 sensors-22-08531-t007:** Quality evaluation-selected articles.

ID	Subject	Author	Score
A01	Assistive Technology	Júnior et al. [81]	4.0
A02	Medicines Recognition	Chang et al. [82]	3.5
A03	Localized Assistive Scene	Ghazal et al. [83]	4.5
A04	Drug Pill Recognition	Chang et al. [84]	5.0
A05	Visually Impaired People	Rao and Singh [85]	2.5
A06	Visually Impaired Pedestrian	Chang et al. [86]	4.5
A07	Pattern Recognition	Bal et al. [87]	3.0
A08	Exploring Printed Text	Su et al. [88]	5.0
A09	Intelligent Navigation	Yadav et al. [89]	5.0
A10	Rehabilitation of People	Jacob et al. [13]	5.0
A11	Visually Impaired Users	Jiang et al. [90]	3.0
A12	Sign Language Recognition	Li et al. [91]	2.5
A13	Sign Language Recognition	Punsara et al. [92]	2.5
A14	Assistive Sign Language	Boppana et al. [93]	5.0
A15	Smart Wheelchair	Al Shabibi and Kesavan [94]	4.0
A16	Personal Assistant	Javed and Sarwar [95]	5.0
A17	Visual Aiding System	Kandoth et al. [96]	3.5
A18	Assistance of Patients	Sharma et al. [97]	5.0
A19	Parkinson’s Disease Assist	Baby et al. [98]	3.0
A20	Assistive Device	Wang et al. [99]	3.5
A21	Navigation System	Kumar et al. [100]	3.0
A22	Sign Language Interpretation	Lee et al. [101]	5.0
A23	Visual Assistive	Sreeraj et al. [102]	4.0
A24	Visual Assistant	Hengle et al. [103]	5.0
A25	Zebra Crossing Detection	Akbari et al. [104]	5.0
A26	Scene-to-Speech Mobile	Karkar et al. [105]	4.5

By the authors.

**Table 8 sensors-22-08531-t008:** Data extraction form.

ID	Field	Values	Objectives
PD 1	ID	Incremental Numeric Value	Study Identification
PD 2	Title	Textual Value	Study Identification
PD 3	DOI	Textual Value	Study Location
PD 4	Machine-learning Model	Textual Value	Answer QP1
PD 5	Topics	Textual Value	Answer QP2
PD 6	Key Words	Textual Value	Answer QP2
PD 7	IoT Device	Textual Value	Answer QP3
PD 8	Addressed Issue	Multiple selection options: hearing impairment, cognitive, impairment, motor impairment, visual impairment, and degenerative disease	Answer QP4

By the authors.

**Table 9 sensors-22-08531-t009:** Algorithms and machine-learning techniques applied, and the respective deficiency addressed for each study.

Applied ML Models	Articles	Impairments
ANN	A1, A10, A13, A21	Visual, motor coordination, hearing
CNN	A6, A8, A9, A11, A12, A14, A24	Visual, hearing, degenerative
RNN	A18, A22	Degenerative, auditory
Multiple CNN	A25	Visual
Clever CNN	A5, A17	Visual
R-CNN	A3, A4	Visual, elderly care
Faster R-CNN	A2, A23	Elderly care, visual
PNN	A7	Hearing
Multi-trained DL models	A26	Visual
Linear regression	A29	Degenerative
SVM	A20, A24	Motor coordination, visual
Independent component analysis	A20	Motor coordination
Naïve Bayes	A16, A18	Cognitive, degenerative
Hoeffding tree	A16	Cognitive
Logistic regression	A16	Cognitive
Random forest	E16	Cognitive
K-means	E16	Cognitive
HOG	E25	Visual

By the authors.

**Table 10 sensors-22-08531-t010:** Topics and occurrences in selected articles.

Topics	Primary Articles
Scene to speech	A1, A3
Assisted navigation	A3, A5, A6, A9, A17, A21, A25
Sign recognition	A7, A12, A13, A14, A22
Object recognition	A2, A4, A9
Object detection	A11, A15, A23
Facial recognition	A21, A24
OCR	A7, A24
Assisted locomotion	A15
Speech recognition	A16
Text to speech	A24
Image captioning	A24
Text detection	A24
Smart assistant	A24
Human activity recognition	A16, A18
Rehabilitation	A10
Self-balancing object	A19

By the authors.

**Table 11 sensors-22-08531-t011:** IoT devices and their occurrences in the selected articles.

IoT Devices	Primary Articles
Portable device	A1, A2, A4, A5, A6, A7, A9, A13, A14, A17, A19, A23, A24
Wearable	A2, A4, A5, A6, A7, A11, A13, A20, A22, A24
Various sensors	A3, A9, A13, A15, A16, A18, A19, A21, A22
Smartphone	A3, A4, A5, A13, A16, A21, A26
Cane	A6, A17
Finger worn wireless	A8
Exoskeleton	A10
Wheelchair	A15
Other	A18
Non-defined	A12, A25

By the authors.

**Table 12 sensors-22-08531-t012:** RaspberryPY, Arduino, and Nvidia Jetson-based IoT devices.

Board	Primary Articles
RaspberryPY	A1, A4, A5, A7, A9, A13, A14, A17, A23, A24
Arduino	A8, A15, A21, A23
Nvidia Jetson	A2, A4

By the authors.

## Data Availability

Not applicable.

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
