# Peer review of "Artificial Intelligence of Things Applied to Assistive Technology: A Systematic Literature Review"

_sensors, 2022, doi:10.3390/s22218531_

Round 1

Reviewer 1 Report

The paper is focused on presenting a literature review of machine learning methods/tools/applications employed for the empowerment of disable individuals. The manuscript is well prepared but not what I have seen in computer vision. In a conventional literature review in computer vision, authors try to summarize the representative studies, identify where the current state of the art stands and what are the future directions. It is rarely states as to how many articles were initially chosen, out of which how many duplicates occurred and what resources are consulted and at what date the search inquiry is conducted. I think authors should revise the paper somewhat to discuss some of the representative articles at length and what are the common themes in those. Here are couple of literature review or survey examples I am familiar with — general from computer vision problems:

Below, I have noted down few other concerns that should be addressed.

In first paragraph, various statistical claims have been made. Are all of the claims stem from reference [1]. In that case it should be clarified. 

On line 41, ‘hearing aids’ is mentioned twice.

Authors should clarify IoT also stands for Internet of Things and may be provide a reference to discriminate the topic at hand i.e., Intelligence of Things in the very beginning of the article. I was unclear until the second paragraph of section 2, where authors finally clarifies between Internet of Things and Artificial Intelligence of Things.  

Line 74: should it be ‘TA’ or ‘AT’ i.e., Assisted Technology. 

Reference missing on line 89. 

Deep neural networks are basically based on convolutional operations compared to the shallow or classic learning methods. I do not consider the number of layers being three or more has to do with deep learning. I think last paragraph of section two should be rephrased to avoid any false claims.

Author Response

Dear Reviewer,
We thank you for your time and availability in reviewing our article, in the attachment we send you a letter detailing all the adjustments made.

Thanks again!

Reviewer 2 Report

Assistive technology is increasingly recognized and is a field of good applicability to artificial intelligence (AI) in addition to universal architecture. AI can be applied in Assistive Technology for people with disabilities. So, this literature review on the topic is very much important at this time. 

(1) The Introduction Section need some extensive work to mention the State-of-the-arts and contributions of the researches clearly. 

(2) Can authors explain a bit about use of Industry 4.0 w.r.t Assistive Technology?  Is inclusion of AI extends the level of the uses of digital technologies? 

(3) As this is a systematic literature review so inclusion of 2-3 use cases will help the readers to understand it well.

Author Response

(The authors gave the same response as above.)

Round 2

Reviewer 1 Report

While authors have addressed some of my minor concerns, the major concerns are still not addressed. In my opinion a reader does not care how many papers were reviewed, how many were duplicate and which resources were used. I keep my point of focusing more on the actual details of representative works instead of listing useless statistics. I keep my original rating of Major Revision, and let the Editor decide if he/she wants to over-rule me. Thanks!

Author Response

Dear Reviewer,

We greatly appreciate your comments and comments to improve our article. Attached we send a letter detailing all the changes made.

The authors.
